# Graphene Oxide Carboxymethylcellulose Nanocomposite for Dressing Materials

**DOI:** 10.3390/ma13081980

**Published:** 2020-04-23

**Authors:** Maria Luisa Saladino, Marta Markowska, Clara Carmone, Patrizia Cancemi, Rosa Alduina, Alessandro Presentato, Roberto Scaffaro, Dariusz Biały, Mariusz Hasiak, Dariusz Hreniak, Magdalena Wawrzyńska

**Affiliations:** 1Department of Biological, Chemical and Pharmaceutical Sciences and Technologies (STEBICEF), University of Palermo, Viale Delle Scienze Bld. 16-17, I-90128 Palermo, Italy; claracarmone@gmail.com (C.C.); patrizia.cancemi@unipa.it (P.C.); valeria.alduina@unipa.it (R.A.); alessandro.presentato@unipa.it (A.P.); 2Institute of Low Temperature and Structure Research, Polish Academy of Sciences, Okólna 2, PL-50-422 Wrocław, Poland; m.markowska@intibs.pl (M.M.); d.hreniak@intibs.pl (D.H.); 3Carbonmed Spółka z Ograniczoną Odpowiedzialnością, ul. Okólna 2, 50-422 Wrocław, Poland; 4Department of Engineering, University of Palermo, Viale Delle Scienze Bld. 6, I-90128 Palermo, Italy; roberto.scaffaro@unipa.it; 5Division of Preclinical Research, Faculty of Health Sciences, Wroclaw Medical University, Ludwika Pasteura 1, PL-50-367 Wrocław, Poland; 6Department of Mechanics and Material Science Engineering, Wrocław University of Science and Technology, Smoluchowskiego 25, PL-50-370 Wrocław, Poland; mariusz.hasiak@pwr.edu.pl

**Keywords:** graphene oxide nanocomposite, carboxymethyl cellulose, biocompatibility, medical devices

## Abstract

Sore, infected wounds are a major clinical issue, and there is thus an urgent need for novel biomaterials as multifunctional constituents for dressings. A set of biocomposites was prepared by solvent casting using different concentrations of carboxymethylcellulose (CMC) and exfoliated graphene oxide (*Exf*-GO) as a filler. *Exf*-GO was first obtained by the strong oxidation and exfoliation of graphite. The structural, morphological and mechanical properties of the composites (CMCx/*Exf*-GO) were evaluated, and the obtained composites were homogenous, transparent and brownish in color. The results confirmed that *Exf*-GO may be homogeneously dispersed in CMC. It was found that the composite has an inhibitory activity against the Gram-positive *Staphylococcus aureus*, but not against Gram-negative *Pseudomonas aeruginosa*. At the same time, it does not exhibit any cytotoxic effect on normal fibroblasts.

## 1. Introduction

Nowadays, there is an urgent need for new innovative solutions in the field of biomedicine, since non-healing wounds remain a major clinical issue and are a significant burden to the medical system [1,2]. Chronic wounds are usually more prone to occur in those patients with a pre-existent physical condition (i.e., diabetes mellitus), and as such can negatively affect the overall clinical outcome. Besides, one of the most important factors that hinders the healing process is represented by bacterial infections. Moreover, the inadequate management of infected wounds is a major cause of amputation or life-threatening sepsis [3]. Thus, an ideal wound dressing should be highly biocompatible, with good mechanical properties and chemical structure, and also show a dynamic role in the wound healing process and be able to prevent bacterial infections [4]. Despite the recent progress in wound management, bacterial infections still lead to significant mortality and morbidity. Therefore, there is a great need for designing advanced materials that could be used as wound dressings. One of the approaches consists of embedding antibiotics within the dressing material [2,5,6,7,8].

Recently, the study of graphene and its derivatives as a promising material for biomedical application—and particularly for wound dressing—has been considered [6,7,8,9,10]. Graphene is a single layer of sp^2^ hybridized carbon atoms creating a hexagonal lattice, and possessing unique properties, i.e., high specific surface area (~2630 m^2^/g), high electron mobility [11,12], thermal conductivity of between 2–5 kW·(m·K)^−1^, [13,14] and the highest known Young modulus (~1 TPa) [15,16]. Moreover, several studies showed that graphene exhibits some antibacterial properties [17,18,19,20]. Its antibacterial mechanism is due to both physical (direct contact of the sharp edge of graphene with bacterial membranes) and chemical (induction of the oxidative stress generated by charge transfer) perturbation. One of the most studied graphene derivatives for bio-applications is graphene oxide (GO), mainly as it is naturally hydrophilic and stable in physiological solutions, and its preparation and functionalization is low-cost and easy to scale-up [21]. GO can be obtained by the oxidation and exfoliation of graphite. In the oxidation process, hydroxyl, carboxyl and epoxide groups are generated on the surface, increasing the GO hydrophilic character and further widening its possible functionalization, and therefore are also proposed for drug delivery and other biomedical applications [22,23,24,25,26,27,28,29,30]. Although GO exhibits antibacterial properties, aggregation phenomena limit its surface area and mode of action. Cellulose is a natural polysaccharide polymer composed of D-glucose units linked together, which can be found in the cell walls of plants and algae, exhibiting properties such as biodegradability, biocompatibility and low cytotoxicity [31]. Thus, cellulose represents a very attractive material for several biomedical applications, including membranes for dialysis, encapsulating agent for drug delivery [32], endotoxin encapsulation and removal, biodegradable implants, dressings for wound treatment, and in-bone regeneration and tissue engineering scaffolds [33].

Carboxymethylcellulose (CMC) is an anionic water-soluble biopolymer, which due to its characteristics such as hydrophilicity, bioadhesivity, pH-sensitivity, non-toxicity, high water solubility, non-allergenicity, low immunogenicity and gel-forming properties, is used in applications such as coating fluids, binders, textiles, paper, food, drug delivery systems and cosmetics [34]. CMC is a semi-synthetic derivative of cellulose produced by partial substitution of hydroxyl groups with carboxymethyl ones in positions 2, 3 and 6. Composite materials based on cellulose and GO—the latter as a filler—were proposed for some medical applications, as such a combination may overcome limitations of cellulose, improving its mechanical and biocompatibility properties [34]. Nor Hazwan et al. developed a reduced GO hydrogel which inhibited biofilm formation by *S. aureus* and *P. aeruginosa*-associated with infected wounds [7]. Rasoulzadeh et al. investigated a CMC/graphene oxide nanocomposite hydrogel beads as a drug delivery system, which is strongly dependent on the pH [35]. Justin et al. demonstrated that GO improves the mechanical properties of chitosan, enhancing its drug delivery profiles [36]. Shao et al. showed that the GO composites possess good cytocompatibility and excellent anti-bacterial rates to *Escherichia coli* and *S. aureus* [20].

In this work, we report the preparation and the characterization of CMC composites containing exfoliated GO (CMCx/*Exf*-GO), also focusing on the evaluation of its biological activity, in terms of both cytotoxicity and antibacterial properties.

## 2. Materials and Methods

### 2.1. Materials

Graphite powder, 11 microns of 99% purity, was purchased from AlfaAesar (Haverhill, MA, USA). Sodium carboxymethyl cellulose (CMC) was purchased from Sigma-Aldrich (Saint Louis, MO, USA), average Mw ~250,000, degree of substitution 0.7. H_2_SO_4_ (98%, p.a.), H_2_O_2_ (30%, p.a.), KMnO_4_ (p.a.), P_2_O_5_ (p.a.) and K_2_S_2_O_8_ (p.a.) were purchased from Chempur, (Piekary Śląskie, Poland), while HCl (35–38%, p.a. basic) was purchased from POCH (Avantor Performance Materials Poland S.A. (formerly POCH), Gliwice, Poland)

### 2.2. Preparation of Exfoliated Graphite Oxide (Exf-GO)

*Exf*-GO was prepared through oxidation of graphite powder in a two-step synthesis (pre-oxidation and actual oxidation), according to the modified Hummers’ method [31,36]. In the first step, a concentrated solution containing H_2_SO_4_ (1.5 mL), K_2_S_2_O_8_ (0.5 g) and P_2_O_5_ (0.5 g) was mixed at 80 °C for 30 min. Then, 1 g of graphite was added. When the color of the mixture became dark blue, it was thermally isolated by coating with aluminum foil and being left to cool at room temperature (6 h). The mixture was then carefully diluted with distilled water, and then filtered and washed with water until the pH became neutral. The solution was air-dried at room temperature for 16 h on top of a Petri Dish, in order to obtain pre-treated graphite.

In the second step, the pre-treated graphite was oxidized. For this purpose, 1 g of the pre-treated graphite powder was added to H_2_SO_4_ (23 mL) cooled in an ice-water bath. KMnO_4_ (3 g) was then gradually added to the mixture, which was maintained at a temperature below 20 °C. After this, the reaction mixture was heated up to 35–40 °C and stirred for 2 h. The mixture was then diluted by adding 46 mL of distilled water. The rapid reaction was stopped after 15 min by adding 140 mL of water and 2.5 mL of 30% H_2_O_2_ solution. The product was then filtered and washed with 1:10 water:HCl solution (250 mL), in order to remove metal ions. The mixture was suspended in distilled water to give a viscous brown 2 wt% dispersion.

Exf-GO was obtained by liquid-phase exfoliation assisted by ultrasonication. Fully oxidized graphene oxide (GO) (0.1 g) was dispersed in distilled water (20 mL). The suspension was sonicated (Ultrasonic dispergator UZDN-A Ukrospribor, Sumy, Ukraine) and purified by centrifugation at 3000 rpm for 30 min (Hermle Universal Centrifuge Z 306, Wehingen, Germany).

### 2.3. Nanocomposites Preparation (CMCx/Exf-GO)

CMC-based nanocomposites have been obtained by solvent casting. Four aqueous dispersions of CMC at 1, 2, 4 and 6 wt% were added to 2.5 mL of *Exf*-GO aqueous dispersion, which were maintained in agitation at room temperature for 3 h and then sonicated for 1 h. The total volume was 30 mL. The pH of each dispersion was 7. The dispersions were then transferred into Petri dishes and kept in an oven at 50 °C to facilitate the evaporation of the solvent, which occurred within 24 h. All the obtained nanocomposites were homogeneous and brown. The colorless and transparent sample of *Exf*-GO free polymer has been also prepared as a reference (Figure 1).

### 2.4. Characterization Techniques

Raman Spectroscopy. Raman spectra were recorded in the 1000–1800 cm^−1^ range using a Renishaw InVia Raman spectrometer equipped with a confocal DM 2500 Leica optical microscope (Wotton-under-Edge, Great Britain, UK), a thermoelectrically cooled CCD as a detector, and an argon laser operating at 514 nm, power 5% (~0.5 mW), acquisition time 10 s and 10 accumulations.

Fourier Transform Infrared Spectroscopy (FT-IR, Billerica, MA, USA). The Attenuated Total Reflectance (ATR) spectra (FT-IR, Billerica, MA, USA) were recorded in the 40–4000 cm^−1^ range, with a step of 2 cm^−1^, by using a FT-IR Bruker Vertex 70 Advanced Research Fourier Transform Infrared Spectrometer (FT-IR, Billerica, MA, USA) equipped with Platinum ATR (diamond crystal).

Atomic Force Microscopy (AFM). Topography of the produced materials (measured as deflection of the cantilever in the vertical direction) was investigated in contact mode by using an Atomic Force Microscope (XE-100, Park Systems, Suwon, Korea). The deflection of the cantilever in the horizontal direction was also measured with the help of a Lateral Force Microscope (NX-100, Park Systems, Suwon, Korea). All measurements were performed for the following scanning area 45 µm × 45 µm, 20 µm × 20 µm, 10 µm × 10 µm, 5 µm × 5 µm, 2.5 µm × 2.5 µm and 1 µm × 1 µm. The data were analyzed using XEI Software (Version 4.3.4) provided by the microscope’s manufacturer.

X-ray Diffraction (XRD). XRD patterns were acquired by a PANalytical X’Pert pro X-ray powder diffractometer (Panalytical, Eindhoven, the Netherlands) using nickel-filtered Cu Kα1 radiation operating at 40 keV and 30 mA.

Mechanical properties. Tensile mechanical measurements were carried out using a dynamometer (model 3365; Instrom, Norwood, MA, USA) on rectangular-shaped specimens (10 × 90 mm) that were cut off from films. The grip distance was in all cases 30 mm, and the crosshead speed 5 mm/min. Five samples were tested for each material.

### 2.5. Biological Activity

Antibacterial tests. *Pseudomonas aeruginosa* ATCC^®^ 10145^™^ and *Staphylococcus aureus* ATCC^®^ 25923^™^ were used as tester strains of Gram-negative and Gram-positive bacteria, respectively. Bacterial cells were pre-cultured for 16 h in sterile glassware tubes containing 3 mL of Luria Bertani (LB) broth (Condalab, Lennox) at 37 °C with shaking (160 rpm). The day after, bacterial cells were inoculated (1% *v*/*v*) in 3 mL of LB broth amended with 2 × 2 cm of different films (composite and polymer film alone reference sample), as well as 10% GO powder in an aqueous solution. Bacterial cells were challenged for 24 h at 37 °C with shaking (160 rpm). The effect of films and *Exf*-GO on bacterial growth was evaluated through the spot plate count method [37], incubating an aliquot of challenged bacterial suspensions on LB agar recovery plates at 37 °C for 24 h, and comparing the number of colony forming units (CFU) to those obtained in the case of an unchallenged culture. The data are expressed as the average of the logarithm of the CFU per milliliter of culture (Log_10_ CFU mL^−1^) for each biological trial (n = 3) with standard deviation.

#### 2.5.1. Fluorescence Microscopy to Assess Loss of Bacterial Membrane Integrity

*P. aeruginosa* and *S. aureus* cells were grown for 24 h in the presence of CMC, CMC6/*Exf*-GO and *Exf*-GO, prior to their imaging by means of a fluorescence microscope (Carl Zeiss, Oberkochen, Germany), as described by [38]. Briefly, bacterial cells were diluted 10 times in phosphate buffer saline (PBS) and stained with a 1:1 (*v*/*v*) mixture of acridine orange (100 μg/mL) and ethidium bromide (100 μg/mL), and were then immediately imaged at 1000× magnification by using excitation and emission wavelength of 488 nm and 550 nm, respectively.

#### 2.5.2. Cell Cultures

IMR-90 (ATCC^®^ CCL-186™) cell line, derived from the normal lungs of a 16-week female fetus and used as a prototype of normal fibroblasts cells, was cultured in Eagle’s Minimum Essential Medium (EMEM), supplemented with 10% fetal bovine serum, 100 U mL^−1^ penicillin and 100 mg mL^−1^ streptomycin. Cells were maintained at 37 °C in a humidified atmosphere of 5% CO_2_, as previously described [39,40,41,42,43,44]. For the treatment, 50,000 cells per well in six-well plates were plated and incubated for 48 h at 37 °C in a CO_2_ incubator. 300 µl of *Exf*-GO, diluted in culture medium, and 4 × 4 cm of each composite were added to the wells for 48 h. At the end of treatment, the cells were observed under inverted microscopy, and then detached from the plates and counted after staining with trypan blue dye.

## 3. Results and Discussion

### 3.1. Structural, Morphological and Mechanical Characterization

The obtained nanocomposites were analyzed to investigate their structure, antibacterial properties and biocompatibility. Raman spectra of exfoliated graphene oxide (*Exf*-GO) and carboxymethylcellulose (CMC) composites containing exfoliated graphene oxide (GO) (CMCx/*Exf*-GO) nanocomposites are reported in Figure 2A, showing two broad overlapping bands in the range of 1200–1800 cm^−1^.

The G band, centered at 1580 cm^−1^, originates from the plane vibrations of carbon rings, while the D band at 1350 cm^−1^ is defect-related, which can be ascribed to carbon-oxygen bonds created during graphite oxidation, grain boundaries and point defects, to name a few, which can cause deviation from the ideal planar structure [42]. The D peak is present in defective carbon materials. The G band is due to the sp^2^ C-C bond stretching in graphitic materials, while the D band is caused by disorder in the graphene structure [45]. The broadened G and D bands observed in all samples indicated the severe disruption of sp^2^ carbon lattice. The position of the two bands was red-shifted respective to the ones of the *Exf*-GO in the Raman spectra of the nanocomposites. This aspect could be due to the structural change in the *Exf*-GO caused by physical or chemical interaction with the CMC. The I_D_/I_G_ ratio, reported in Figure 2B, relates to the sp^3^/sp^2^ carbon ratio that is used to estimate the degree of defects. The observed trend indicates that all composites have a lower ID/IG ratio than that of *Exf*-GO, suggesting that the *Exf*-GO structure in the composites is less organized and the defects decreased with increasing concentrations of CMC. On the contrary, Yadav et al. observed an increase of the I_D_/I_G_ ratios when GO in embedded in the CMC nanocomposites, which suggested that the structure of CMCn/*Exf*-GO nanocomposite was of the ordered carbon nanosheet type [44].

Infrared (IR) spectra and X-ray diffraction (XRD) patterns were recorded to investigate possible interactions between *Exf*-GO and CMC. All IR spectra and XRD patterns of nanocomposites are reported in the Appendix A. Here, representative IR spectra and XRD patterns of CMC6*/Exf*-GO and CMC 6% film are reported (Figure 3A,B).

The typical IR spectrum of CMC shows a broad absorption band at 3440 cm^−1^, due to the –OH group stretching. The bands at 2924 and 2856 cm^−1^ are due to C–H asymmetric and symmetric stretching of the CH_2_ vibration. The bands at 1590, 1415, 1324 and 1053 cm^−1^ are due to bending modes of the –COO group, the scissoring of –CH_2_, the bending vibration of –OH and the stretching of the CH–O–CH_2_ group, respectively. No significant differences, in terms of IR bands (both shape and position), are observed in the IR spectra of the nanocomposites, suggesting that *Exf*-GO does not interact with CMC through intermolecular hydrogen bonds. By adding *Exf*-GO, however, the C–H stretching modes of CH_2_ vibration became sharper as two partially overlapped bands, indicating that there should be good compatibility between CMC and GO, as evident by the transparency of the material.

The XRD pattern of *Exf*-GO shows a strong peak at 2 = 10.35° and a large band at 2 = 20.18°, corresponding to d-spacing of 8.54 and 4.40 Å respectively, which are related to the inter-layer distance between the GO layers and to a short-range order in stacked graphene layers. The XRD patterns of CMC films show two large bands (101 and 110 planes of cellulose–direction c of the unit cell is along the chain axis of the polymer), indicating a relative ordered structure characterized for the crystalline form of cellulose II [45,46,47,48]. The position of the peaks does not significantly change with CMC concentration. They fall at 2 = 11.96° and 20.18°, corresponding to d-spacing of 7.39 and 4.40 Å, respectively. Regardless of the CMC amount (i.e., 1, 2, or 3%), the presence of *Exf*-GO does not cause any change in the nanocomposite. Alternately, when combined with CMC 6%, the disappearing of the band can be observed at 2 = 11.96°, while the second band at 20.18° remains almost unvaried (see Figure 3B). The band of *Exf*-GO is also not present. These results indicated a crystal transition of cellulose during the composite formation in the presence of *Exf*-GO, indicating that *Exf*-GO was well exfoliated in the CMC matrix, being finely dispersed in the latter, in agreement with Raman data. The glyosidic linkages between the sugar units were partially destroyed during the dissolving and embedding of *Exf*-GO, which resulted in a probable reduction of the crystallinity [7].

Atomic force microscopy (AFM) can provide spatial information both parallel and perpendicular to the surface, thus allowing a 3D reconstruction of the sample surface. In doing so, we investigated both the topographical properties and the quality of the materials. 2D/3D AFM topography images of all CMC films and nanocomposites are reported in the Appendix A of the Support Information. For a general comment, here we report those of CMC6/*Exf*-GO and CMC 6% film (Figure 4).

AFM images showed that all films displayed a typical regular morphology. Analysis of the topography images obtained for the investigated materials shows that no visible grains were observed on the surface of the tested CMC and CMCx/*Exf*-GO samples. The same conclusion was obtained from simultaneous analysis of the topographies during left-to-right and right-to-left scanning, with the help of later force microscopy (LFM). Roughness Rq, Ra and Rz for the AFM measured topographies of CMC and CMCx/*Exf*-GO composites (Figure 5), calculated with the help of XEI software (Park Systems), suggest that the surface’s irregularities for all investigated materials result only from the preparation process of both CMC and CMCx/*Exf*-GO materials. This is particularly visible for the CMC 2% sample, for which the roughness parameter Rz is higher than for other materials. On the other hand, the roughness values presented in Figure 5 clearly suggest excellent quality of both CMC and CMCx/*Exf*-GO materials. The preparation of Exfoliated Graphite Oxide samples perfectly reflects the surface of CMC films, and the roughness of the CMCx/*Exf*-GO composite is not significantly different with respect to CMC films.

In order to study the effect of *Exf*-GO on the tensile properties of the CMC, the tensile test was carried out. Representative stress-strain curves of CMC6/*Exf*-GO nanocomposites and CMC 6% film are reported in Figure 6. Adding *Exf*-GO to CMC causes an increase of the elastic modulus, causing it to rise from 65 MPa to about 80 MPa, i.e., +23%. The increase in tensile stress is even more impressive, as it rises from about 45 MPa to about 80 MPa, i.e., +78%. This could be due to the different organization of the CMC chains in the composite [49,50]. The elongation at break, however, seems almost unaffected by the presence of the filler. It is worth noting that this set of mechanical properties is compatible with the applications foreseen for these materials.

The antibacterial and the cytotoxic activities were evaluated on CMC 6%-based composites, the composite constituted by CMC 6%, and on 2.5 mL *Exf*-GO (so called CMC6/*Exf*-GO), since this formulation, among those investigated, showed the lower value I_D_/I_G_, and is also the lesser organized.

### 3.2. Antibacterial and Cytotoxic Activity

*S. aureus* cell growth was inhibited by the presence of either CMC or CMC6/*Exf*-GO in the culture broth, which determined a decrease in the number of viable cells, corresponding to 0.8 and 1 log units (Figure 7), respectively. On the other hand, none of the analyzed samples affected the growth of *P. aeruginosa*, reaching 3.7 ± 1.6 × 10^9^ CFU mL^−1^ under each experimental condition (Figure 7). These results are in line with other studies aimed to unveil the antibacterial potential of CMC-based formulations, such as those containing reduced graphene oxide (rGO) [7] sodium alginate and chitosan [51], a zinc-based metal-organic framework and graphene oxide [52], or sodium alginate and pyrogallic acid [53], all of which in the form of either composite or bio-nanocomposite films. Here, a reasonable explanation for the lack of the antimicrobial activity of the composite films against the *P. aeruginosa* strain may rely on the higher resistance of Gram-negative bacteria as compared to Gram-positive ones, due to differences in the structure of the cell envelope. Indeed, Gram-positive strains are protected by a cell wall mostly composed of mucopeptide, while Gram-negative microorganisms possess a thin layer of the latter and an additional layer of lipoproteins and lipopolysaccharides (LPS) [54,55], which might contribute to a high degree of resistance towards the composites tested in this study. Antibacterial activity against Gram-positive bacteria is much more frequent than that one against Gram-negative bacteria ones [42,56,57,58,59,60]. The effect exerted by either CMC or CMC6/*Exf*-GO upon *S. aureus* and *P. aeruginosa* cells was corroborated by fluorescence microscopy imaging, after dual staining with acridine orange and ethidium bromide (Figure 8). *S. aureus* cells treated with CMC and CMC6/*Exf*-GO appeared swollen and emitted red fluorescence deriving from ethidium bromide dye, likely due to a loss of cell membrane integrity, in contrast to untreated and GO exposed cells, which appeared to be vital. Alternately, *P. aeruginosa* cells were vital, as they mostly emitted green fluorescence due to the acridine orange dye in all the tested conditions. All the above test results clearly indicate that the obtained composite can be considered for use as a bioactive advanced material for wound dressings against infections due to Gram-positive bacteria.

GO, CMC and CMC6/*Exf*-GO nanocomposite were also evaluated for their biocompatibility, performing cytotoxicity tests on normal fibroblasts cell line (IMR-90). As a result, no cytotoxic effect was observed on normal fibroblasts after 48 h of treatment. CMC6/*Exf*-GO was more resistant to solubilization than CMC, although its structural change in terms of softening and delamination was observed (Figure 9A). Moreover, none of the tested samples inhibited cell growth (Figure 9B), and no morphological changes were observed under optical microscopy (Figure 9C). These results support the conclusion that the synthesized composites are biocompatible, and suitable for biomedical applications.

## 4. Conclusions

Nanocomposites consisting of carboxymethylcellulose (CMC) and exfoliated graphene oxide (*Exf*-GO) were prepared and characterized by using microstructural techniques and biological tests. Raman, infrared (IR) spectra and X-ray diffraction (XRD) patterns showed that there is an increasing of disorder in the *Exf*-GO into polymer structure, as well as in the CMC chain structure. Atomic force microscopy (AFM) images evidenced no changes in the morphology of the CMC in presence of *Exf*-GO. The AFM and later force microscopy (LFM) investigations of the topography show a good quality of both CMC and *Exf*-GO samples. Moreover, the fabrication process of CMCx/*Exf*-GO materials does not change topography-obtained materials. On the other hand, the increase in tensile stress was observed in the composite compared to CMC.

It was demonstrated that the composite constituted by CMC 6% and on 2.5 mL *Exf*-GO exhibits antimicrobial activity against Gram-positive *S. aureus*. At the same time, the same composite does not exhibit any cytotoxic effect on normal fibroblasts and haemolytic activity, which makes it a promising candidate for use in medical materials such as wound dressing.

## Figures and Tables

**Figure 1 materials-13-01980-f001:**
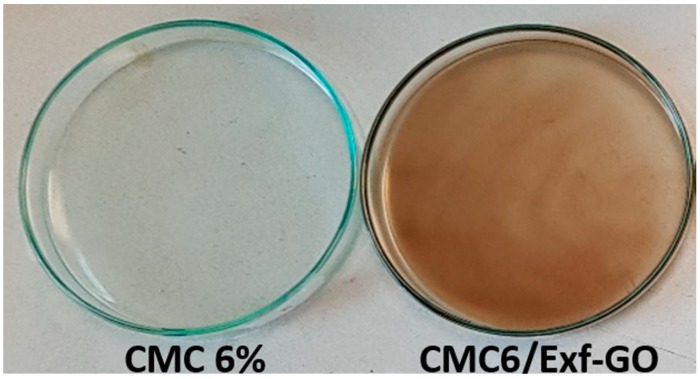
Photo of the CMC 6% film and CMC6/*Exf*-GO nanocomposite in Petri dishes.

**Figure 2 materials-13-01980-f002:**
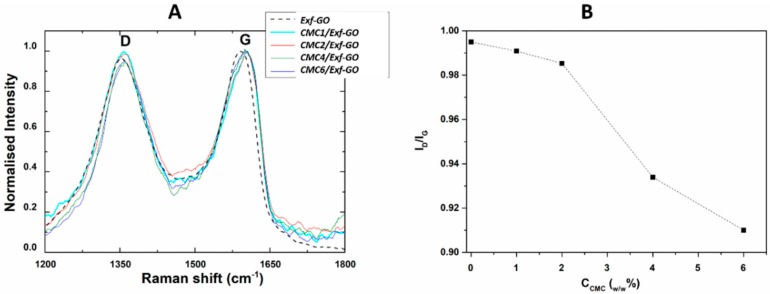
(**A**) Raman spectra of CMCx/*Exf*-GO nanocomposites; (**B**) I_D_/I_G_ ratio for CMC-based nanocomposites as a function of CMC%.

**Figure 3 materials-13-01980-f003:**
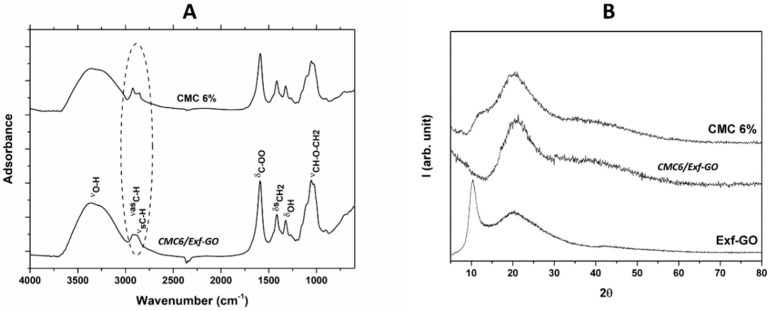
(**A**) IR spectra of CMC6*/Exf*-GO nanocomposites and CMC 6% film; (**B**) XRD patterns of *Exf*-GO, CMC6/*Exf*-GO and CMC 6% film.

**Figure 4 materials-13-01980-f004:**
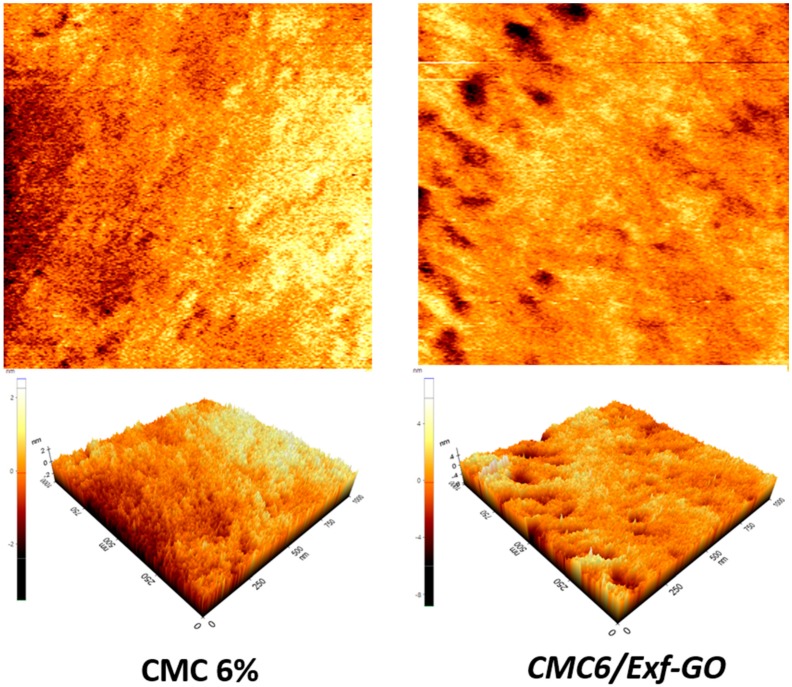
2D/3D AFM topography images for the CMC films (left side) and CMC6/*Exf*-GO nanocomposite (right side) obtained in contact mode for a scanning area of 1 µm × 1 µm.

**Figure 5 materials-13-01980-f005:**
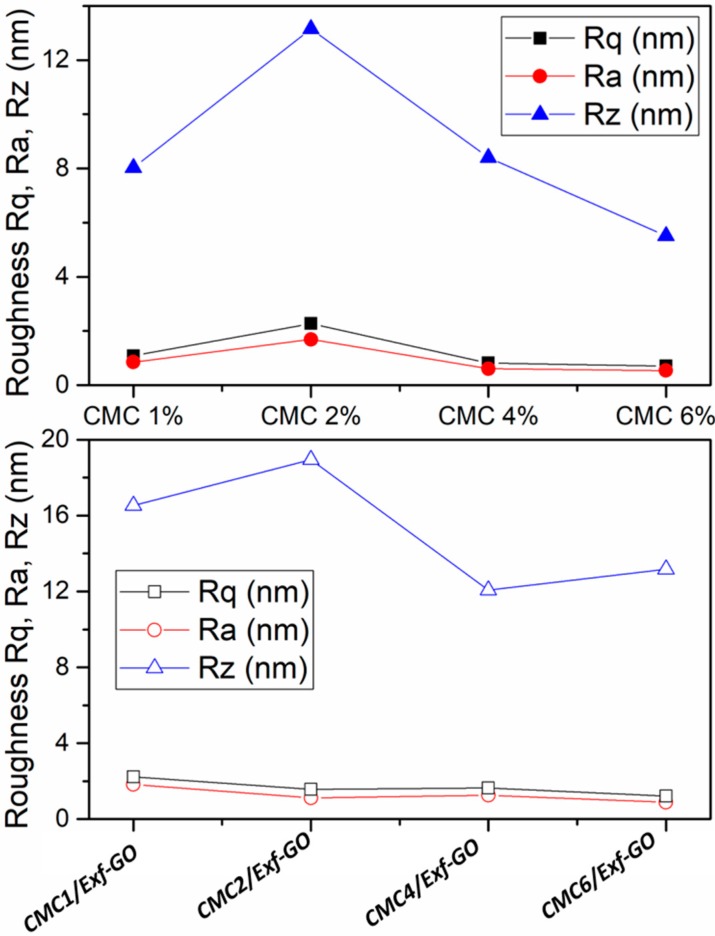
Roughness Rq, Ra and Rz for the CMC samples (upper figure, close symbols) and CMC6/*Exf*-GO samples (lower figure, open symbols), where: Rq is the root-mean-squared roughness; Ra is the roughness average; and Rz is the ten-point average roughness.

**Figure 6 materials-13-01980-f006:**
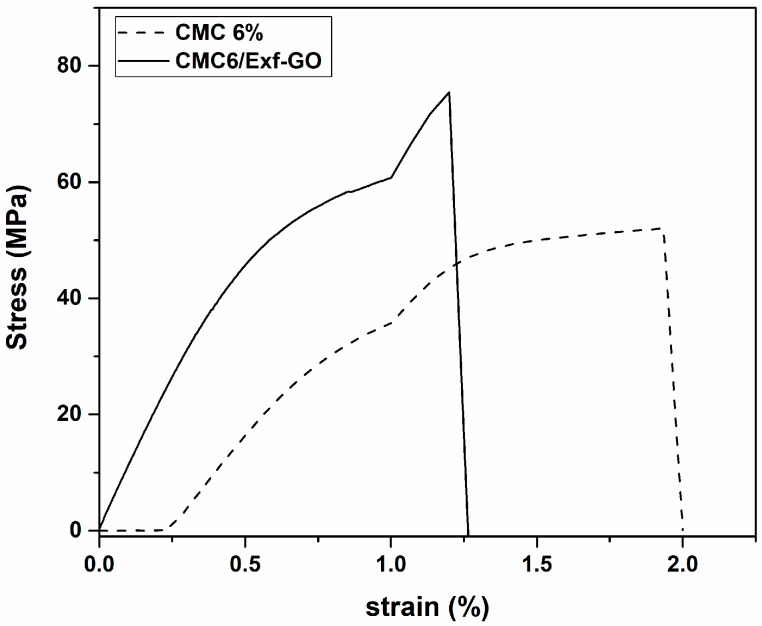
Stress-Strain curves of CMC6/*Exf*-GO nanocomposites and CMC 6% film.

**Figure 7 materials-13-01980-f007:**
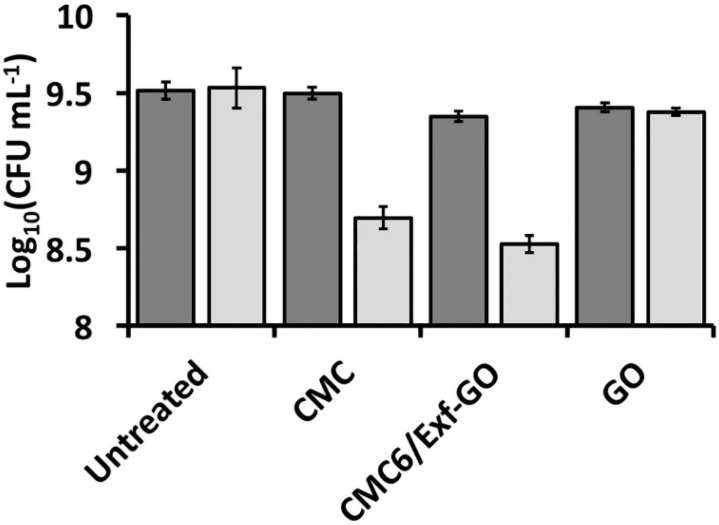
Bar graph reporting the biomass yield of both *P. aeruginosa* ATCC^®^ 10145^™^ (dark grey bar) and *S. aureus* ATCC^®^ 25923^™^ (light grey bar).

**Figure 8 materials-13-01980-f008:**
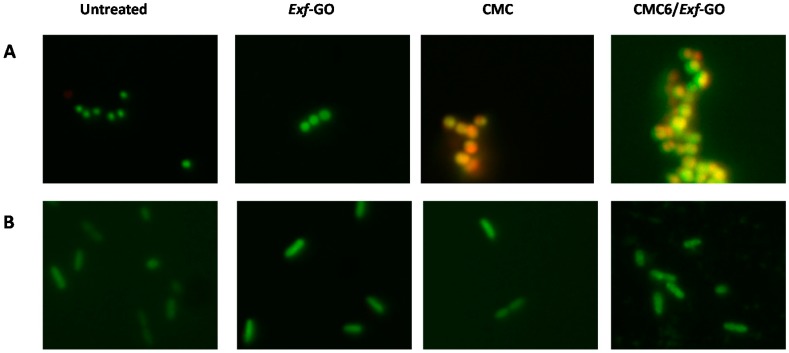
Fluorescence microscopy images of *S. aureus* ATCC^®^ 25923^™^ (**A**) and *P. aeruginosa* ATCC^®^ 10145^™^ (**B**), either not exposed (untreated) or exposed to *Exf*-GO, CMC and CMC6*/Exf*-GO.

**Figure 9 materials-13-01980-f009:**
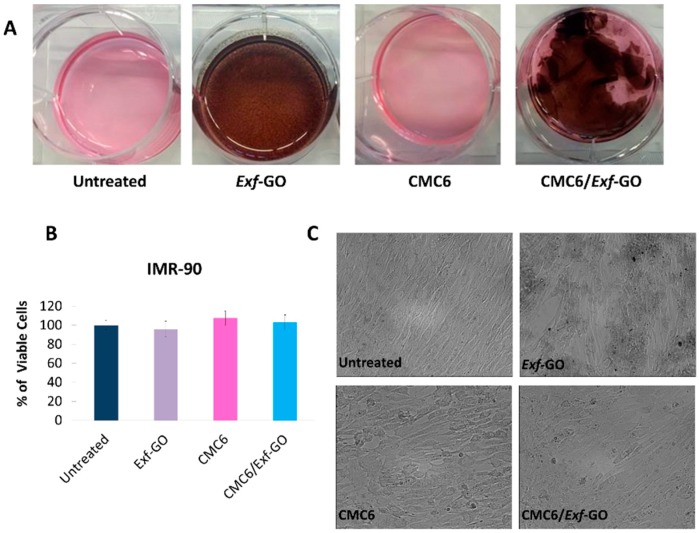
The effect of CMC6/*Exf*-GO treatment on normal fibroblasts IMR-90 cell line; (**A**) details of the culture plates in which the fibroblasts not exposed (untreated) or exposed to *Exf*-GO, CMC and CMC6/*Exf*-GO were grown; (**B**) graph showing the % of viable cells with respect to untreated ones, counted by trypan blue exclusion method; and (**C**) optical microscopy images of normal fibroblasts either not exposed (untreated) or exposed to GO, CMC and CMC6/*Exf*-GO for 48 h, where no morphological changes were observed. Magnification 200×.

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
