# Peer review of "Graphene Oxide Carboxymethylcellulose Nanocomposite for Dressing Materials"

_materials, 2020, doi:10.3390/ma13081980_

Round 1
Reviewer 1 Report
In this manuscript entitled “Graphene oxide carboxymethylcellulose nanocomposite for dressing materials” the author has introduced and investigated a composite film based on carboxymethylcellulose (CMC) and exfoliated graphene oxide as a nano-filler for using wound healing application. I believe that the findings of the manuscript are not significant and also the represented data is poor. Moreover, the methodology must be explain more in detail and results discussion should be analysis clear hypotheses about the data are tested. The discussion section is often considered the most important part of research but in this manuscript, the discussion part is too poor. Furthermore, the manuscript is not well written with numerous grammatical mistakes and confusing sentences. Most importantly, the plagiarism of the article is about 30%, which is not acceptable in academic environment. I believe that the novelty of the work is also not enough to be published in this journal. Overall, the manuscript suffers from shortcomings, and therefore I recommend that is rejected.
Author Response
Reviewer #1:
In this manuscript entitled “Graphene oxide carboxymethylcellulose nanocomposite for dressing materials” the author has introduced and investigated a composite film based on carboxymethylcellulose (CMC) and exfoliated graphene oxide as a nano-filler for using wound healing application.
I believe that the findings of the manuscript are not significant and also the represented data is poor.
Authors do not agree with this opinion. The research is highly interdisciplinary, correlates data from different fields of interest related to structural, mechanical and biological properties of the nanocomposite material for the important application. In the revised version authors did the best to emphasize more the importance of this work.
Moreover, the methodology must be explain more in detail and results discussion should be analysis clear hypotheses about the data are tested.
Authors started in their experimental work from the thesis, not hypothesis. It is stated clearly: we propose the use of obtained material for the particular application (sore wound dressings) Therefore, performed research was focused on developing the easy method of preparation and the most important desired properties pointed out in the Introduction: “Thus, an ideal wound dressing should be highly biocompatible, with good mechanical properties and chemical structure, showing also a dynamic role in wound healing process and able to prevent bacterial infections”. Methodology presented in the paper is directly following the aims indicated in the last part of introduction: “…focusing also on the evaluation of its biological activity, in terms of both cytotoxicity and antibacterial properties.”
The discussion section is often considered the most important part of research but in this manuscript, the discussion part is too poor.
The authors agree with the opinion that discussion (should always) be considered as the most important part of research. In contrast, the authors do not agree with the opinion that it is too poor in our work. We have tried to discuss the data obtained in the best possible way to maintain an interdisciplinary approach, not going into too much detail and presenting and discussing the data that is most relevant for the aims our work. We plan to develop more detailed research in further, more specialised studies.
Furthermore, the manuscript is not well written with numerous grammatical mistakes and confusing sentences.
We did our best to check once again the entire manuscript to find and correct both English and the remaining misprints.
Most importantly, the plagiarism of the article is about 30%, which is not acceptable in academic environment. I believe that the novelty of the work is also not enough to be published in this journal.
The Reviewer should be more careful writing such bad opinion without referring any source data, references etc. - it is not common way of reviewing for any scientific journal. Authors are experienced researchers and such statement is really disrespectful for them. Using a special software (CrossRef Similarity and others) for checking the manuscript is a great opportunity for editors and reviewers but the obtained statistics should be analysed very carefully and critically. Authors being also editors, reviewers and authors are using the same software to check their manuscripts before submission and are sure their originality. If any sentence about the methodology used (reported in the experimental part) is similar to ones reported in other papers it is obvious because the instrumentations and methodologies have been already reported by the authors in their earlier publications. They are not part of the original research (as the quotes, bibliography) and therefore they should be excluded. Once again: presented research and its presentation is novel and as declared by all co-authors never published before. Our manuscript meets the highest requirements of the Journal.
Overall, the manuscript suffers from shortcomings, and therefore I recommend that is rejected.
We are leaving this decision to Editor and other Reviewers.
Reviewer 2 Report
1.What kind of cell is IMR-90 cell line? Are the normal fibroblasts from the description in Figure 9? After 48 hours of incubation, the cells are fully attached, but whether there are quantitative results to confirm that there are significant differences between the groups
2.In Fig 9(A), If the black part represents the Exf-GO, the Exf-GO in the composite film (CMC6/Exf-GO) should be non-uniformly distributed.
3.What standard method is used to perform the hemolysis test? Please list references or standard numbers. On the other hand,Is it possible to quantify by colorimetry?
Author Response
Dear Reviewers,
thank you for all your comments to our manuscript. I have carefully read all the comments and the answers are listed below.
Author’s reply:
- IMR 90 cells derived from normal lungs of a 16-week female fetus and were used as prototype of normal fibroblasts cells. This information was added in the MM section. As stated in MM section, after incubation with treatments cells were detached from the plates and counted after trypan blue staining. A new graph showing the % of viable cells with respect to untreated cells was added in figure 9. No significant differences in cell viability were observed between the treated cells, suggesting that the composite does not exhibit any cytotoxic effects on normal fibroblasts.
- In Figure 9, the Exf-GO powder precipitated on cell wells after 48h of treatment, as possible to observe in the Figure 9C micrograph, where a brown precipitated is visible on the Exf-GO treated cells. On the contrary, in the nanocomposite, Exf-GO is incorporated within the CMC.
- As suggested by Reviewer 1, the hemolytic activity was removed since it did not give any additional information to the manuscript.
Please find the revised document ( the corrections are distinguished with red colour) attached herewith and the correcetd version of Fig.9.
Yours sincerrely,
Dariusz Biały,
MD,PhD, SCi
Reviewer 3 Report
Graphene oxide carboxy methyl cellulose nanocomposite for dressing materials
Comments
1. What is the novelty of this work?
2. Line 22-24, check sentence making, repetition of word prepared; Line 102 “To this
purpose”, should be “for this purpose”; Line 116 “were added of” should be “were
added to”; Line 117, “shacked”?
3. Why 10% GO powder in aqueous solution was used for antibacterial activity? How
the concentration was decided?
4. Which cell line is IMR-90? Detail should be added and why this cell line was
selected?
5. Hemolytic activity of wound dressing is needed?
6. IR of plain ExfGO should have been added for better understanding of compatibility.
7. What can be the reason for increase in tensile strength of nanocomposite?
8. Cytotoxicity data is not clear.
9. What is the status of bioadhesiveness of the prepared nanocomposite?
10. this paper can be cited https://doi.org/10.3390/biom9080363
Author Response
Dear Reviewers,
thank you for all your comments to our manuscript. I have carefully read all the comments and the answers are listed below.
Author’s reply:
- This work reports the study on CMC/Exf-GO nanocomposites trying to combine the structural and morphological data with the biological activity. As stated in the introduction “Recently, the study of graphene and its derivatives, as a promising material for biomedical application and in particular for wound dressing, has been considered [6-10].” Our paper can be considered a more deepen study respect to the other similar studies already published.
- The English and Editorial errors were corrected. All the changes in the revised form of the manuscript are highlighted in red.
- The quantity GO is the same in all nanocomposites. As it is underlined in the raw 276 “The antibacterial and the cytotoxic activities were evaluated on CMC6% based composites the composite constituted by CMC 6% and on 2.5 mL Exf-GO (so called CMC6/Exf-GO) since this formulation, among those investigated, showed the lower value ID/IG and it is the less organized.
- IMR 90 cells derived from normal lungs of a 16-week female fetus and used as prototype of normal fibroblasts cells. This information was added in the MM section.
- Hemolytic activity was removed.
- The purpose of the IR investigation was to study the CMC features in presence of ExfGO. We acquired the IR spectra by using platinum ATR (diamond crystal). Since the refractive index of the crystal is the same of the carbon based materials like the ExfGO, we cannot acquire the spectra of ExfGO by using this ATR.
- The increase in tensile strength of nanocomposite can be due both the interaction between the two components (van der walls and p-p) and the different organization of the CMC chains in the composite. The following sentence was added in the raw 271“It could be due to the different organization of the CMC chains in the composite.”.
- Cytotoxicity test was performed to assess the biocompatibility of nanocomposite. As, expected, no significant differences in cell viability were observed between the treated cells, suggesting that the composite does not exhibit any cytotoxic effects on normal fibroblasts. A new graph, showing the % of viable cells with respect to untreated cells was added in figure 9. Please see the revised manuscript.
- We did not evaluated the bioadhesiveness of the prepared nanocomposite
- The suggested paper was added in the manuscript as ref 31.
Yours sincerrely,
Dariusz Biały,
MD,PhD, SCi
Round 2
Reviewer 1 Report
I'm satisfied with the authors corrections. For example they said "If any sentence about the methodology used (reported in the experimental part) is similar to ones reported in other papers it is obvious because the instrumentations and methodologies have been already reported by the authors in their earlier publications. ", The authors need to consider about plagiarism not to change the plagiarism concept. There is no 'another' author in this case. These are your words, your language, your thoughts, and your expressions. You may have used them twice, which would constitute dual publication, but that is not the same as self-plagiarism.
- Publication of the same script more than once (actual duplicate publication)
- Publication of several scientific products based on (partially) the same material or disseminating (partially) the same results (overlapping publications)
- Re-use of own text, structure, ideas, interpretations, etc. (self-plagiarism)
Moreover, there is not only plagiarism in the materials and method but also there is in the discussion or introduction texts, for examples;
"Both D and G peaks are the result of vibrations of sp2-bonded carbon atoms. The G peak is a result of in-plane vibrations of sp2 bonded carbon atoms, whereas the D peak is due to out of plane vibrations attributed to the presence of structural defects." it has exactly copy pasted from this article from page 7921: Layek, Keya, and Kalyan Kumar Mistry. "A Study of Optimization of Various Parameters in the Fabrication of Screen-Printed Electrodes." IEEE Sensors Journal 18.19 (2018): 7917-7923.
Or
"Among graphene derivatives, graphene oxide (GO) has been widely explored, taking advantage of its high solubility and stability in physiological solutions, cost-effectiveness, scalable production, and facile biological and chemical functionalization" from this artcile: Bayer, Ilker S. "Thermomechanical properties of polylactic acid-graphene composites: a state-of-the-art review for biomedical applications." Materials 10.7 (2017): 748. even they did not cite this article in here.
Therefore, beside the novelty of this work and other concerns, I do not recommend for accepting and publishing this article.
Author Response
Replies were sent to the editor
Reviewer 3 Report
The authors have addressed the reviewer's comments.
Author Response
Thank you for the comments. I attach the revised version of the manuscript. Sincerely D. Biały